# Knowledge gaps and acquisition about HPV and its vaccine among Brazilian medical students

**Annielson de Souza Costa[1], Jéssica Menezes Gomes[1], Ana Cláudia Camargo Gonçalves Germani[2], Matheus Reis da Silva[1], Edige Felipe de Sousa Santos[3], José Maria Soares Júnior[1], Edmund Chada Baracat[1], Isabel Cristina Esposito Sorpreso[1]***

**1** Disciplina de Ginecologia, Departamento de Obstetricia e Ginecologia, Faculdade de Medicina FMUSP, Universidade de São Paulo, São Paulo, SP, Brazil, **2** Departamento de Medicina Preventiva, Faculdade de Medicina FMUSP, Universidade de São Paulo, São Paulo, SP, Brazil, **3** Study Design and Scientific Writing Laboratory at ABC Medical School, Santo André, SP, Brazil

* icesorpreso@usp.br

**Data Availability Statement:** The dataset is available on the Harvard Dataverse, https://doi.org/10.7910/DVN/IR3UJO.

## Abstract

### Objective

To analyze factors associated with knowledge gaps and acquisition about HPV and its vaccine among medical students.

### Method

Cross-sectional and analytical study conducted at the University of São Paulo Medicine School, in 2016. A convenience sample of students completed a data collection instrument containing questions on knowledge about HPV and its vaccine, and vaccine acceptability. The level of knowledge and acceptability established as a "good level" was 80% of correct answers on the questionnaire. Internal validity was calculated with Cronbach's alpha value (α) = 0.74. Bivariate and multiple analyzes were performed using the Stata® program (Stata Corp, College Station, USA) 14.0.

### Results

To evaluate the internal consistency of the instrument applied, the Cronbach's alpha equation was used, obtaining the alpha value (α) = 0.74 for this population. This value attests that the consistency of the answers obtained with this questionnaire is considered substantial and acceptable. Among the 518 medical students who completed the survey, the majority were men 312 (60.4%) with a mean age of 23 (± 2.8) years old; 199 (38.3%) of the students were in the final years of graduation (5th and 6th years). Students in the first, second and third year of study had a 51% higher risk of a knowledge gap when compared to students in the final years of graduation [PR 1.51 (1.3:1.8); p <0.001]. Men were at 22% higher risk of unsatisfactory knowledge than women are [PR 1.22 (1.07: 1.39). There was no knowledge acquisition during medical school in the following questions (p <0.05), indication of vaccine for individuals with HIV and contraindication in pregnant patients.

**Funding:** The author(s) received no specific funding for this work.

**Competing interests:** The authors have declared that no competing interests exist.

## Conclusion

Male medical students, in the first year of medical school, and those who were not vaccinated had significant knowledge gaps about HPV. The novelty of the study includes the finding of non- acquisition of knowledge during the medical school graduation on safety and vaccination schedule and vaccine administration in specific populations.

## Introduction

Knowledge acquisition about HPV, its clinical repercussions, and its vaccine is developed throughout the graduate years of medical education in developed and developing countries [1,2]. The skills and competences acquired by these students will be used for men's and women's health care, especially for HPV-induced cancer (cervical, oropharynx, anus, rectum and penis) and precursor lesions at different levels of care, including health counseling and education, prevention, diagnosis, treatment and recovery [2,3].

As future health care providers, these medical students will perform preventive actions (HPV vaccination) and promotion of cervical cancer screening [4], which is a high prevalence and mortality disease in Brazil and worldwide, representing a public health problem [5,6]. In Brazil the HPV vaccine is available in the National Immunization Program for female population aged nine to 14 years old, male population aged 11 to 14 years old [7].

The attitude of non-recommendation of HPV vaccine by health professionals is related to their own understanding of HPV and the vaccine [8]. Berenson et al. (2015) [9] found that senior medical students had insufficient basic knowledge about HPV epidemiology and HPV vaccine.

The identification of knowledge gaps and barriers of acceptance during medical education can support proposals for improving the content delivered to students, reduce negative beliefs, and promote HPV vaccines recommendation [10]. Thus, medical students should be encouraged to receive education about HPV, prevention and vaccination, to enable them to deliver effective vaccine recommendations, guiding their patients and accurately responding to questions [9–11]. The objective of this research was to analyze factors associated with knowledge gaps and acquisition about HPV and its vaccine among medical students.

## Method

### Study design, place and date

This was a cross-sectional study conducted at the University of São Paulo Medicine School, São Paulo, Brazil, 2016.

### Participants

The population was the 900 students enrolled in the Medicine School of the University of Sao Paulo. The appropriate sample size for the study, based on parameters [12] of 5% error, 95% confidence level, and 80% test power was estimated to 270 individuals. A convenience sample of both sexes' students over the age of 18 from the first to sixth year of medical school were invited to participate voluntarily in the research, whit no financial incentive offered to the respondents, who signed the Informed Consent Form.

**Non-inclusion criteria.** Individuals with cognitive problems or any interest that may influences the answer to the questions contained in the data collection instrument.

## Data collection procedure

Data collection was performed by undergraduate medical students previously trained to administer the instruments with the support and supervision of the researcher. To minimize selection bias, data collection occurred in student classrooms, between classes, on breaks in clinical rotations or in housing units. After completing the instrument, accurate information about HPV was provided and questions answered.

## Instrument

The instrument (Box 1) is a previous study developed and applied in similar populations. In addition to socio-demographic questions, there were 31 questions divided into six sections [13,14]:

---

### Box 1. Instrument: HPV knowledge and acceptability questionnaire

**Domain 1:** Knowledge about HPV

**1. Do you know what HPV is?**

( )no  ( )yes ( )not sure

**2. Is HPV a virus?**

( )no  ( )yes ( )not sure

**3. Is HPV a sexually transmitted disease?**

( )no  ( )yes ( )not sure

**4. Can HPV cause cervical cancer?**

( )no  ( )yes ( )not sure

**5. Can HPV cause changes in the Pap (screening for cervical cancer)?**

() no () yes () not sure

**6. Is cervical cancer a major cause of cancer in women?**

( )no  ( )yes ( )not sure

**7. Can smoking increase the risk of cervical cancer?**

( )no  ( )yes ( )not sure

**Domain 2: Knowledge about HPV vaccine**

**8. Does the HPV vaccine prevent cervical cancer?**

( )no  ( )yes ( )not sure

**9. Should the HPV vaccine be given before the first sexual intercourse?**

( )no  ( )yes ( )not sure

**10. Can the HPV vaccine be given to people who have had sex?**

( )no  ( )yes ( )not sure

---

**11. Can HPV vaccine be harmful to health?**

( )no   ( )yes  ( )not sure

**12. Can the HPV vaccine cause HPV infection?**

( )no   ( )yes  ( )not sure

**13. Is the HPV vaccine provided by the Government?**

( )no   ( )yes  ( )not sure

**14. Is the HPV vaccine part of the girls' immunization record?**

( )no   ( )yes  ( )not sure

**15. Where did you hear about the HPV vaccine?**

()School ()Friends ()TV/radio ()Internet ()Health professional ()Others_______

**16. Are 3 doses required for complete vaccination?**

( )no   ( )yes  ( )not sure

**17. Does the HPV vaccine lessen the chance of having genital warts?**

( )no   ( )yes  ( )not sure

**18. Does the HPV vaccine decrease the chance of having Pap (cervical cancer screening) changes?**

( )no   ( )yes  ( )not sure

**Domain 3: HPV Vaccine Barriers**

**19. Do you think the HPV vaccine would stimulate the onset of sexual life earlier?**

( )no   ( )yes  ( )not sure

**20. Do you think that after the HPV vaccine you still need to use a condom?**

( )no   ( )yes  ( )not sure

**21. Do you think that after the HPV vaccine you still need to have the pap (cervical cancer screening)?**

( )no   ( )yes  ( )not sure

**Domain 4: Acceptability of HPV vaccine**

**22. Do you know anyone who has already had the HPV vaccine?**

( )no   ( )yes  ( )not sure

**23. Have you taken the HPV vaccine yet?**

()no ()yes/ If yes ()public services ()particular services ()not sure

**24. Would you recommend the HPV vaccine for a child, friend, or relative to take?**

( )no   ( )yes  ( )not sure

**Domain 5: Personal Background**

**Answer only if you are female**

**25. Have you ever had pap (cervical cancer screening)?**

( )no   ( )yes   ( )not sure

**26. Have you ever had cervical cancer?**

( )no   ( )yes   ( )not sure

**27. Have you ever had genital warts?**

( )no   ( )yes   ( )not sure

**Domain 6: Health professionals**

**Answer if you are a health professional**

**28. Patients living with HIV can take the vaccine?**

( )no   ( )yes   ( )not sure

**29. Am I confident to indicate HPV vaccination for patients?**

( )no   ( )yes   ( )not sure

**30. Do I feel confident giving information about HPV to patients?**

( )no   ( )yes   ( )not sure

**31. Can pregnant women take the vaccine?**

( )no   ( )yes   ( )not sure

The Domain 5: Personal background was not considered in this study as it concerns medical personal information.

## Variables

The dependent variables were gender, age, marital status, number of children, year of medical study, education, income, grade, and vaccination status. The independent variables were the total scores for each of the themes: knowledge about HPV, knowledge about HPV vaccine, barriers to HPV vaccination, acceptability of HPV vaccine and specific knowledge of health professionals.

The good level of knowledge was classified as adequate when the percentage of correct answers exceeded 80% and was based on the previously published score in a similar population [14]. Knowledge acquisition was considered when students in 1st, 2nd and 3th years of medical school achieved less than 80% correct answers and students in 5th and 6th years reached more than 80%. This shows that throughout the course the student gained knowledge about HPV and its vaccine.

## Data analysis

Data was entered into Excel and analysis performed by two researchers (EFSS and ASC). All analyzes were performed using the Stata program (Stata Corp, College Statiom, USA) 14.0. The dataset is available on the Harvard Dataverse, https://doi.org/10.7910/DVN/IR3UJO [15].

Knowledge acquisition exists when the percentage of correct answers in the basic cycle (1st, 2nd and 3rd year of medical study) is less than 80% and in internship cycle (5th and 6th year of medical study) is higher than 80% limit. This shows that throughout the course, the student acquired knowledge about HPV and its vaccine.

For descriptive data analysis, quantitative variables were computed for measures of central tendency (mean, median and quartiles) and dispersion measures (variance, standard deviation and interquartile range), according to the data adherence to Gaussian distribution. And the Shapiro-Wilk test.

Qualitative variables were expressed as absolute and relative frequencies. Confidence intervals of the respective measurements were calculated.

For bivariate analysis of categorical variables, the chi-square test was used and for multiple analysis, Poisson regression was performed. Values were considered statistically significant with two-tailed p-value <0.05 with a 95% confidence interval.

Non-responding was represented as missing and did not exceed 5% of question-by-question responses. The response rate of the variables (questionnaire questions) was an average of 99.2% (SD 0.9%).

## Ethical aspects

The research followed the rules of the National Health Council and was approved by the Faculdade de Medicina–Universidade de Sao Paulo Research Ethics Committee under opinion No. 1,938,072.

## Results

To evaluate the internal consistency of the instrument applied, the Cronbach's alpha equation was used, obtaining the alpha value ($\alpha$) = 0.74 for this population. This value attests that the consistency of the answers obtained with this questionnaire is considered substantial and acceptable [16].

Of the 520 medical students who answered the questionnaire, the majority of respondents were male 60.2% (312), 63.8% (332) aged 20–24 years, 98.9% (489) of the respondents were single and two students reported having children (Table 1).

According to grade 61.7% (321) students were in 1st, 2nd and 3th year of medical study and 38.2% (199) were in 5th and 6th year of medical study. Of the participants, 81.5% (419) were not vaccinated for the HPV vaccine (Table 1).

Table 2 shows the factors associated with the knowledge gap about HPV, its clinical repercussions and its vaccine among students considering the variables: gender, age group, marital status, graduation period, having been vaccinated and income. Multivariate analysis showed a difference in knowledge by gender (p <0.01) and year of study (p <0.01). Men had a 22% higher risk of having unsatisfactory level of knowledge than the women interviewed [PR 1.2 (1.1: 1.4)]. It is observed that there was a difference in students by grade with first, second and third year students having a 51% risk of having a knowledge gap than students in the final cycle [PR 1.51 (1.3:1.8)] (Table 2). Students who were not vaccinated against HPV were 41% more likely to have a knowledge gap than vaccinated students [PR 1.41 (1.12: 1.78)].

Differences in knowledge of HPV and its vaccine between men and women (p <0.05) were identified in the questions "Can smoking increase the risk of cervical cancer?" [0.87 (0.79: 0.97)], "Can the HPV vaccine be given to people who have had sex?" [0.93 (0.87: 0.99)], "Can the HPV vaccine cause HPV infection?" [0.82 (0, 73: 0.92)], "Is the HPV vaccine part of the girls' immunization records?" [0.77 (0.67: 0.89)], "Are 3 doses required for complete vaccination?" [0, 57 (0.49: 0.67)], "Does the HPV vaccine decrease the chance of having changes in the

**Table 1. Medical students sociodemographic characteristics, medical school of the university of Sao Paulo, Brazil, 2016.**

| Characteristics | N (520) | % (CI 95%)* |
|---|---|---|
| **Sex** | | |
| Male | 312/518 | 60.2 (56 : 64) |
| Female | 206/518 | 39.8 (36 : 44) |
| **Age** | | |
| ≤ 19 years old | 46 | 8.8 (6.0 : 12) |
| 20–24 years old | 332 | 63.8 (60 : 68) |
| 25–29 years old | 131 | 25.2 (22 : 29) |
| ≥ 30 years old | 11 | 2.1 (1 : 3) |
| **Marital Status** | | |
| Singles | 489/494 | 99.0 (97 : 99) |
| Stable Union | 5/494 | 1.0 (0.4 : 2.4) |
| **Children** | | |
| Yes | 2/516 | 0.4 (0.1 : 1.5) |
| No | 514/516 | 99.6 (98 : 100) |
| **Year of graduation** | | |
| 1st, 2nd and 3th | 321 | 61.7 (58 : 46) |
| 5th and 6th | 199 | 38.3 (34 : 43) |
| **HPV vaccinated** | | |
| No | 419/514 | 81.5 (78 : 85) |
| Yes | 95/514 | 18.5 (15 : 22) |
| **Family Icome**** | | |
| >R$ 9.745,00 | 255/511 | 49.9 (45 : 54) |
| de R$ 7.475,00 a R$ 9.745,00 | 116/511 | 22.7 (19 : 26) |
| de R$ 1.734,00 a 7.475,00 | 40/511 | 7.8 (5.8 : 10) |
| de R$ 1.085,00 a 1.734,00 | 9/511 | 1.8 (0.9 : 3.3) |
| < R$ 1.085,00 | 91/511 | 17.8 (15 : 21) |

*CI 95%: Confidence Interval of 95%

**1 American dólar ($) = 4,07 Brazilian currency (R$)

Pap smear test?" [0.90 (0.82: 0.98)], "Do you think the HPV vaccine will stimulate the onset of sexual activity at an earlier age?"[0.94 (0.89: 0.98)], (S1 Table).

Table 3 highlights the knowledge acquisition (not acquired / acquired) and the knowledge level (adequate / inadequate) of participants by class level. There was no knowledge acquisition and the level of knowledge was inadequate in the fists years of study in the following questions (p <0.05): "Can smoking increase the risk of cervical cancer?" [0.65 (0.57 : 0.75)]; "Can HPV vaccine be harmful to health? [0.66 (0.53: 0.83)]; "Is the HPV vaccine part of a girls' vaccination card?" [0.75 (0.66 : 0.86)]; "Are 3 doses required for full vaccination?" [0.76 (0.65 : 0.90)]; "Can patients living with HIV get the vaccine?" [0.56 (0.44:0.71)]; "Can pregnant patients get the vaccine?" [0.50 (0.25: 1.00)].

## Suitable knowledge level was considered at 80%

The questions "should HPV vaccine be applied before first sexual intercourse?" And "does HPV vaccine reduce the chance of genital warts?" [0.96 (0.86: 1.07)]; [1.02 (0.89: 1.16)], respectively, did not show significance in the comparison between the basic and hospitalized cycle, however, showed that both groups did not have a good level of knowledge and there was no knowledge acquisition during graduation.

**Table 2. Factors associated with knowledge gap about HPV, its clinical repercussions and its vaccine among medical students at the medical school of university of São Paulo, Brazil, 2016.**

| Variables | Total PR (95% CI)* | P-value |
|---|---|---|
| **Sex** | | |
| Male | 1.0 | |
| Female | 1.2 (1.1 : 1.4) | <0.01 |
| **Age** | | |
| ≤ 19 years old | 1.0 | |
| 20–24 years old | 0.9 (0.6 : 1.2) | 0.41 |
| 25–29 years old | 0.8 (0.6 : 1.0) | 0.11 |
| ≥ 30 years old | 0.8 (0.6 : 1.1) | 0.14 |
| **Marital Status** | | |
| Singles | 1.0 | |
| Stable Union | 0.5 (0.2 : 1.6) | 0.26 |
| **Período de graduação** | | |
| Basic cycle** | 1.0 | |
| Final cycle*** | 1.5 (1.3 : 1.8) | <0.01 |
| **HPV vaccinated** | | |
| No | 1.0 | |
| Yes | 1.4 (1.1 : 1.8) | <0.01 |
| **Family Income †** | | |
| >R$ 9.745,00 | 1.0 | |
| de R$ 7.475,00 a R$ 9.745,00 | 0.8 (0.5 : 1.4) | 0.40 |
| de R$ 1.734.00 a 7.475.00 | 1.1 (0.9 : 1.3) | 0.23 |
| de R$ 1.085.00 a 1.734.00 | 1.0 (0.8 : 1.1) | 0.70 |
| < R$ 1.085.00 | 1.0 (0.8 : 1.1) | 0.65 |

*PR (95%CI): Prevalence Ratio (Confidence Interval of 95%) calculated by Poisson regression.

**Basic cycle: 1st 2nd and 3th graduation year

***Final cycle: 5th and 6th graduation year

† 1 American dólar ($) = 4,07 Brazilian currency (R$)

The main sources of information about HPV and its vaccine shown in Fig 1 and reported by participants were: "Healthcare Professionals" (40%, n = 207), "Media (TV / radio)" (28%, n = 146) and "School" (26% n = 136).

## Discussion

This study analyzed actors associated with the HPV knowledge gap and HPV vaccine acceptability among medical students. Results can support medical education given the importance of the subject in public health and the introduction of the vaccine into Brazil's National Immunization Program in 2014 [17]. We found that males had attended first, second and third year of medical study and had not been vaccinated against HPV are related factors to gap of knowledge.

Knowledge gaps between males and females about HPV and its vaccine has been found in both medical students [18–21] and in the non-university population, corroborating our data [22,23]. While the higher level of knowledge found in females in this study is a function of the personal and individual experiences, since medical students of both sexes are exposed to the same amount of theoretical concepts before graduation [3]. In addition, a study shows that male parents underestimate the effects of HPV in males and prioritize the vaccine for women

**Table 3. Acquisition and level of knowledge about HPV and its vaccine among medical students, according to grade, at the medical school of university of São Paulo, Brazil, 2016.**

| Questions | Grade | | PR (CI95%)*** | p-value | Knowledge Acquisition | Knowledge Level |
|---|---|---|---|---|---|---|
| | 1st, 2nd and 3th year* | 5th and 6th year** | | | | |
| 1. Do you know what HPV is? | 303 (94.4) | 198 (99.5) | 1.9 (0.9 : 1.0) | <0.01 | not acquired | Suitable |
| 2. Is HPV a virus? | 320 (99.7) | 199 (100) | 1.0 (1.0 : 1.0) | 0.32 | not acquired | Suitable |
| 3. Is HPV a sexually transmitted disease? | 193 (97.8) | 301 (93.8) | 0.9 (0.9 : 1.0) | 0.01 | not acquired | Suitable |
| 4. Can HPV cause cervical cancer? | 199 (100) | 315 (98.1) | 1.0 (1.0 : 1.0) | 0.01 | not acquired | Suitable |
| 5. Can HPV cause changes in Pap smear | 276 (86.2) | 199 (100) | 0.9 (0.8 : 0.9) | <0.01 | not acquired | Suitable |
| 6. Is cervical cancer a leading cause of death in women? | 278 (86.6) | 192 (97) | 0.9 (0.8 : 0.9) | <0.01 | not acquired | Suitable |
| 7. Can smoking increase the risk of cervical cancer? | 159 (49.7) | 150 (75.4) | 0.6 (0.6 : 0.7) | <0.01 | not acquired | Not suitable |
| 8. Does the HPV vaccine prevent cervical cancer? | 211 (66.1) | 165 (83.8) | 0.8 (0.7 : 0.9) | <0.01 | adquirida | Suitable |
| 9. Should the HPV vaccine be given before the first sexual intercourse? | 232 (72.3) | 149 (74.9) | 1.0 (0.9 : 1.1) | 0.51 | not acquired | Not suitable |
| 10. Can the HPV vaccine be given to people who have had sex? | 265 (82.5) | 183 (92) | 0.9 (0.8 : 0.9) | <0.01 | not acquired | Suitable |
| 11. Can the HPV vaccine be harmful to your health?* | 101 (31.6) | 94 (47.2) | 0.7 (0.5 : 0.8) | <0.01 | not acquired | Not suitable |
| 12. Can the HPV vaccine cause HPV infection?* | 193 (60.3) | 161 (80.9) | 0.7 (0.7 : 0.8) | <0.01 | Acquired | Suitable |
| 13. Is the HPV vaccine provided by the government? | 258 (80.6) | 175 (88.4) | 1.0 (0.8 : 1.0) | 0.01 | not acquired | Suitable |
| 14. Is the HPV vaccine part of the girls' immunization records? | 172 (53.6) | 140 (70.7) | 0.7 (0.7 : 0.9) | <0.01 | not acquired | Not suitable |
| 16. Are 3 doses required for complete vaccination? | 151 (47.6) | 122 (61.9) | 0.8 (0.7 : 0.9) | <0.01 | not acquired | Not suitable |
| 17. Does the HPV vaccine decrease the chance of having genital warts? | 210 (66.2) | 127 (64.8) | 1.0 (0.9 : 1.2) | 0.74 | not acquired | Not suitable |
| 18. Does the HPV vaccine decrease the chance of having changes in the Pap smear test? | 237 (75.2) | 160 (81.6) | 0.9 (0.8 : 1.0) | 0.84 | Acquired | Suitable |
| 19. Do you think the HPV vaccine will stimulate the onset of sexual activity at an earlier age?* | 290 (91.8) | 184 (93.4) | 1.0 (0.9 : 1.0) | 0.49 | not acquired | Suitable |
| 20. Do you think that you still need to use a condom after HPV vaccination? | 316 (100) | 196 (100) | 1.0 | 1.00 | not acquired | Suitable |
| 21. Do you think that you still need to have a Pap smear test after HPV vaccination? | 314 (99.4) | 196 (100) | 1.0 (1.0 : 1.0) | 0.16 | not acquired | Suitable |
| 22. Do you know anyone who has already received the HPV vaccine? | 190 (59.9) | 119 (60.4) | 1.0 (0.8 : 1.1) | 0.92 | Not applicable | Not applicable |
| 23. Have you received the HPV vaccine yet? | 70 (22.1) | 25 (12.7) | 1.7 (1.1 : 2.6) | 0.01 | Not applicable | Not applicable |
| 24. Would you recommend the HPV vaccine for a child, friend, or relative? | 259 (82) | 186 (94.4) | 0.9 (0.8 : 0.9) | <0.01 | Not applicable | Not applicable |

*(Continued)*

**Table 3.** (Continued)

| Questions | Grade | | PR (CI95%)*** | p-value | Knowledge Acquisition | Knowledge Level |
|---|---|---|---|---|---|---|
| | 1st, 2nd and 3th year* | 5th and 6th year** | | | | |
| **28. Patients living with HIV can take the vaccine?** | 86 (27.4) | 94 (48.4) | 0.6 (0.4 : 0.7) | <0.01 | Acquired / not acquired | Suitable/ Not suitable |
| **29. Am I confident to indicate HPV vaccination for patients?** | 204 (65) | 147 (75.8) | 0.8 (0.8 : 1.0) | 0.01 | Not applicable | Not applicable |
| **30. Do I feel confident giving information about HPV to patients?** | 140 (44.6) | 147 (76.6) | 0.6 (0.5 : 0.7) | <0.01 | Not applicable | Not applicable |
| **31 Can pregnant patients get the vaccine?** | 14 (4.5) | 17 (8.8) | 0.5 (0.2 : 1.0) | 0.05 | Acquired / not acquired | Suitable/ Not suitable |

*Basic cycle: 1st 2nd and 3th graduation year

**Final cycle: 5th and 6th graduation year

***PR (CI 95%): Prevalence ratio (Confidence Interval of 95%) calculated by Poisson regression.

[24]. It is necessary to disseminate correct information to parents and to emphasize the fact that they are not actively involved and open to dialogue when it comes to their children's sexual education [25].

Regarding the undergraduate period, it was evidenced that students in the fists years of graduation have a greater knowledge gap than students in the final cycle. Similar results were found in the research by Yam et al, (2017) [26] and Silva et al, (2017) [27], which showed that senior medical students (3rd grade or above) have a higher level of knowledge than junior medical students (below 3rd grade). This is expected as senior students experience greater exposure to HPV classes and content.

The results show that not having been vaccinated against HPV was associated with lower levels of knowledge. Other studies have found low HPV vaccination rates among medical students was related to lack of knowledge about vaccine safety and efficacy, religious and cultural issues, and lack of government programs that cover the age range of students [28–31]. In addition, vaccination rates are higher in countries with national HPV vaccination programs [3]. Vaccinated students have been found to be more willing to recommend HPV vaccination and clarify patient questions and are more likely to follow the recommendations and take every opportunity to offer HPV vaccine [30,31].

For the question "Can smoking increase the risk of cervical cancer?" there was no knowledge acquisition throughout medical school and there was a knowledge gap between the students in the first and and last study years. Silva et al, (2017) [27] found different results in their research, showing that medical students understand that cervical cancer is associated with behavioral patterns such as smoking. Exposure, age at onset and frequency of cigarette smoking are factors that influence the incidence of cervical intraepithelial neoplasia (CIN) and cervical cancer [32]. The relation between smoking and cervical cancer should be reinforced in the medical school curriculum since smoking is a public health problem and interferes with the health-disease process.

In the question, "Should the HPV vaccine be given before the first sexual intercourse?", Although there was no significant difference in knowledge between students of the basic and final cycle, it was observed that both have insufficient level of knowledge. This theme should be reinforced during the undergraduate years as it can be a barrier to vaccination because it is a sex-related theme and can be taboo in specific populations.

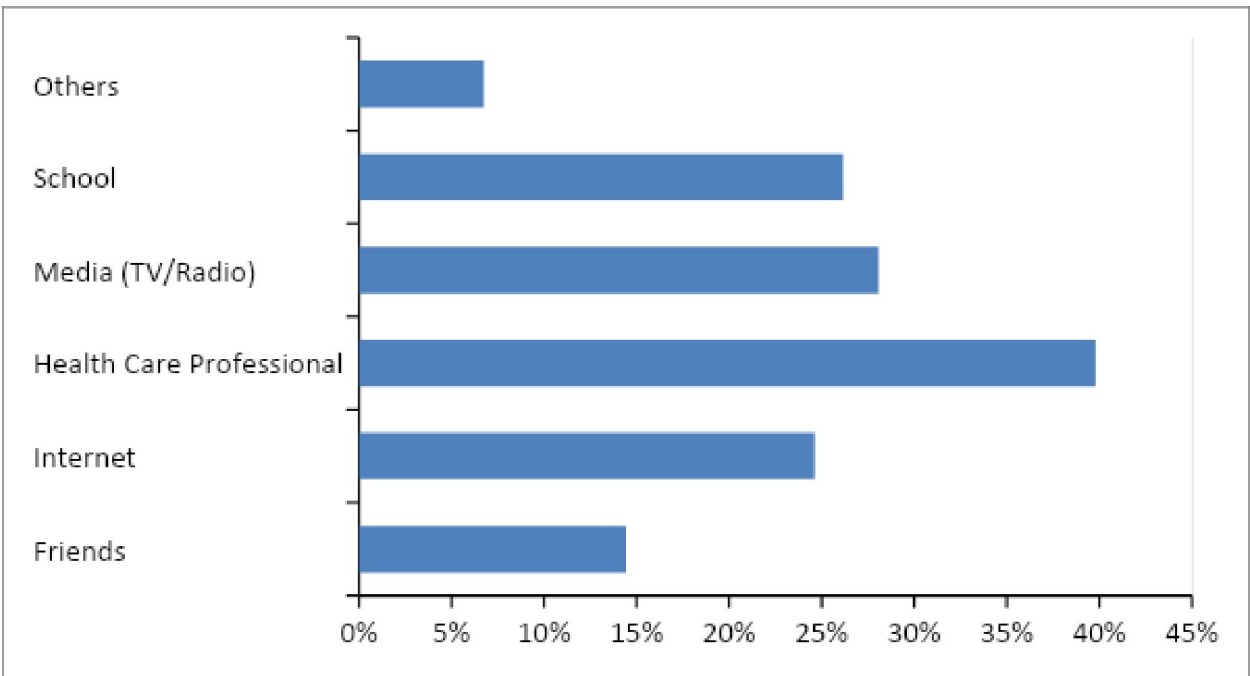

**Fig 1. Sources of information on HPV and its quadrivalent vaccine among medical students.**

There was no acquisition of knowledge along the graduation to the question "The HPV vaccine may be harmful to health?". The vaccine is safe and approved by the World Health Organization Global Vaccine Safety Advisory Board [33]. Vaccine safety needs to be included in the undergraduate subjects, since it is in universities that the professionals responsible for transmitting information to the population are trained and thus become informers of this new technology introduced in the health system.

Programs for successful implementation of HPV vaccination include factors such as intersectoral involvement (across the education and health sector), as well as collaboration between institutions in the health, education and financial sectors. The future efforts should focus on programs that can be implemented within health care settings, such as reminder and recall strategies and physician-focused efforts, as well as the use of alternative community-based locations, such as schools [34,35].

In Brazil, in 2014, vaccine implementation had been used together with the education sector in public and private schools [33,35], bringing benefits in terms of vaccine coverage rates. Our study reinforces the themes that must be addressed during medical training in order to strengthen the physician's action in providing information and patient's counseling. In the questions, "Are 3 doses required for full vaccination?" And "Is HPV vaccine part of the girl's vaccination calendar?" There was no knowledge acquisition during the medical course and a knowledge gap was verified between the students of the basic cycle and the final cycle. The HPV vaccine was implanted in Brazil by PNI in March 2014, and then the vaccination schedule was modified with adjustment in the number of doses required and changes in the target population [19]. During graduation, the importance of reviewing the National Immunization Program and its updates contained in the National Health System (NHS) should be reinforced.

In the question, "Does HPV vaccine decrease the chance of having genital warts?" no knowledge acquisition occurred, and it remained inadequate in both periods. Studies show

substantial effects of HPV vaccine in reducing anogenital warts and precancerous lesions [36–38].

The students in this research have significant knowledge gaps in specific questions to health professionals; as the indication of the vaccine for people with HIV and the contraindication in pregnant patients. Diverging from our results, Silva and Monteiro (2016) [39] identified in another Brazilian region that medical students demonstrated knowledge about the need for cervical cancer screening in HIV-positive women. Studies show that health professionals are important in vaccine counseling [40]. Moreover, the relationship of trust between the health-care professional and the patient exists through clear and accessible and sometimes culturally appropriate communication and information about the HPV vaccination program [41]. This relationship is reflected in "healthcare professionals" being the main sources of information about HPV and its vaccine, followed by "Media (TV / radio)" and "School". Williams et al, (2013) [40] points out that health care providers play a more active role in recommending HPV vaccines, and recent studies show that a patient receiving a doctor's recommendation is 4–5 times more likely to get vaccinated. Our results on information sources reinforce the importance of creating spaces on health topics for medical student self-care.

It is worth mentioning that the phenomenon called vaccine hesitation (which consists of delay in accepting or refusing vaccines, despite the availability of vaccination services) [41,42], is a fundamental determinant for low HPV vaccination rates. Studies show that potential determinants of vaccine hesitation are communication and preventive attitudes. Health professionals act as a key factor in reducing vaccine hesitation [43,44].

Studies [45–47] demonstrate the importance of the physician in primary health care setting and vaccination counseling. Furthermore, the authors describe care practices based on the patient's medical relationship and health promotion interventions based on the individual care. Our study shows information and knowledge gaps that must be acquired during the medical graduation so that the student can exercise health education and individual counseling of his patients.

Limitations of the study include the cross-sectional design which does not allow understanding over time how and when individuals gained knowledge about HPV, it is not possible to distinguish whether knowledge acquisition was a cause or a consequence according to the student's level of education, neither the individual acquisition of each student. The instrument did identify other possible associated factors such as ethnicity, culture and religion that may be associated. The data collection took place in an internationally recognized University by convenience sample and findings may not be representative of other Brazilian medical schools.

Integration between health professionals and the population is key to ensuring adequate vaccine coverage and ensuring promising vaccine results. The encouragement of information, counseling and continuing education is recommended as a strategy to broaden the acceptance of the vaccine in order to settle its implementation and ensure its effectiveness in reducing future cervical cancer cases [41,42].

Thus, our study brings the novelty of the non-acquisition of knowledge during medical school, especially among unvaccinated males on the important issues in women's and men's health. Other areas of minimal knowledge is the relationship between tobacco use and cervical cancer, the safety of vaccine, schedule and schedule used by the NHS, being allowed to administer the vaccine in HIV patients and not indicated among pregnant women.

## Conclusion

Female and students in 5th and 6th years are more knowledgeable about HPV and its vaccine. Males, the basic cycle and not being vaccinated are factors associated with knowledge gaps among medical students.

The gaps in knowledge acquisition during graduation occur on the themes: relationship of tobacco with cervical cancer, safety and vaccination scheme in special populations such as immunosuppressed and pregnant women should be reinforced in the contents of medical training programs.

## Supporting information

**S1 Table. Knowledge about HPV and its quadrivalent vaccine among medical students by gender, medical school of university of São Paulo, Brazil, 2016.**
(PDF)

## Author Contributions

**Conceptualization:** Annielson de Souza Costa, Edmund Chada Baracat, Isabel Cristina Esposito Sorpreso.

**Data curation:** Annielson de Souza Costa, Jéssica Menezes Gomes, Matheus Reis da Silva, Edige Felipe de Sousa Santos, Isabel Cristina Esposito Sorpreso.

**Formal analysis:** Jéssica Menezes Gomes, Matheus Reis da Silva, Edige Felipe de Sousa Santos, José Maria Soares Júnior.

**Investigation:** Annielson de Souza Costa.

**Methodology:** Jéssica Menezes Gomes, Ana Cláudia Camargo Gonçalves Germani, Matheus Reis da Silva, Edige Felipe de Sousa Santos, José Maria Soares Júnior, Isabel Cristina Esposito Sorpreso.

**Project administration:** Annielson de Souza Costa, José Maria Soares Júnior, Edmund Chada Baracat, Isabel Cristina Esposito Sorpreso.

**Resources:** Annielson de Souza Costa.

**Supervision:** Edige Felipe de Sousa Santos, Edmund Chada Baracat, Isabel Cristina Esposito Sorpreso.

**Validation:** Ana Cláudia Camargo Gonçalves Germani, Isabel Cristina Esposito Sorpreso.

**Visualization:** Annielson de Souza Costa, Jéssica Menezes Gomes, Ana Cláudia Camargo Gonçalves Germani, Edmund Chada Baracat, Isabel Cristina Esposito Sorpreso.

**Writing – original draft:** Annielson de Souza Costa, Jéssica Menezes Gomes, José Maria Soares Júnior.

**Writing – review & editing:** Annielson de Souza Costa, Jéssica Menezes Gomes, Ana Cláudia Camargo Gonçalves Germani, Matheus Reis da Silva, Edige Felipe de Sousa Santos, José Maria Soares Júnior.

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
