## [Decision Letter · Decision Letter 0]

3 Jan 2020

PONE-D-19-33466

KNOWLEDGE GAPS AND ACQUISITION ABOUT HPV AND ITS QUADRIVALENT VACCINE AMONG MEDICAL STUDENTS

PLOS ONE

Dear Dr. Esposito Sorpreso,

Thank you for submitting your manuscript to PLOS ONE. After careful consideration, we feel that it has merit but does not fully meet PLOS ONE’s publication criteria as it currently stands. Therefore, we invite you to submit a revised version of the manuscript that addresses the points raised during the review process.

/span>

We would appreciate receiving your revised manuscript by February 5, 2020. To enhance the reproducibility of your results, we recommend that if applicable you deposit your laboratory protocols in protocols.io, where a protocol can be assigned its own identifier (DOI) such that it can be cited independently in the future. For instructions see: http://journals.plos.org/plosone/s/submission-guidelines#loc-laboratory-protocols

We look forward to receiving your revised manuscript.

Kind regards,

Italo Francesco Angelillo, DDS, MPH

Academic Editor

PLOS ONE

Journal Requirements:

2. Please clarify in your Methods section whether the questionnaire is published under a CC-BY license, or whether you obtained permission from the publisher to reproduce the questionnaire in this manuscript. Please explain any copyright or restrictions on this questionnaire

3. Thank you for including your ethics statement:

"The research followed the rules of the National Health Council and was approved by

the Research Ethics Committee under opinion No. 1,938,072.".

i) Please amend your current ethics statement to include the full name of the ethics committee/institutional review board(s) that approved your specific study.

ii) Once you have amended this/these statement(s) in the Methods section of the manuscript, please add the same text to the “Ethics Statement” field of the submission form (via “Edit Submission”).

4. Please include your tables as part of your main manuscript and remove the individual files. Please note that supplementary tables (should remain/ be uploaded) as separate "supporting information" files

6. Your ethics statement must appear in the Methods section of your manuscript. If your ethics statement is written in any section besides the Methods, please move it to the Methods section and delete it from any other section. Please also ensure that your ethics statement is included in your manuscript, as the ethics section of your online submission will not be published alongside your manuscript.

Reviewers' comments:

Reviewer's Responses to Questions

**Comments to the Author**

1. Is the manuscript technically sound, and do the data support the conclusions?

Reviewer #1: Partly

Reviewer #2: Partly

2. Has the statistical analysis been performed appropriately and rigorously? 

Reviewer #1: Yes

Reviewer #2: No

3. Have the authors made all data underlying the findings in their manuscript fully available?

Reviewer #1: Yes

Reviewer #2: No

4. Is the manuscript presented in an intelligible fashion and written in standard English?

Reviewer #1: No

Reviewer #2: No

5. Review Comments to the Author

Reviewer #1: This paper examines factors associated with knowledge about HPV infection, related diseases and vaccine among medical students at the University of São Paulo, Brazil.

I am afraid that I have to recommend rejecting the present manuscript as it stands. I detail below what my reasons for doing so are and encourage the authors to improve their manuscript.

First of all, and I think this is the main problem of the paper, I noticed substantial grammar/language/syntax mistakes in the manuscript and tables. I kindly suggest you to have your paper proof read by a native speaker, if necessary multiple times, before you submit. Moreover, there is a lack of clear contribution in the present manuscript.

Title

I suggest to delete in the title and in the introduction the word “quadrivalent”. The survey questions generically explored HPV vaccine.

Introduction

In the first paragraph, the authors stated “The skills and competences acquired by these students will be used for women's health care…” Long-lasting infections with high-risk HPVs can cause cancer in different parts of the body, such as oropharynx, anus, rectum, penis, and not only in the cervix, vagina, and vulva. In developed countries, high-risk HPVs cause 3% of all cancers in women and 2% of all cancers in men. Please, clarify that the competences acquired by medical students will be also useful to protect men's health care.

I suggest to add a brief description of what the Brasil’s National Immunization Program recommend regarding HPV vaccination.

Methods

The authors described how the sample size was calculated, but the reference used for sample size calculation to estimate the prevalence was not included.

It would seem that “did not accept” to participate in the study was an ineligibility criterion, but the number of subjects who refused to answer the survey, represents a fraction of people in the total sample.

In the Instrument paragraph, no. 6), and throughout the manuscript, I suggest to replace health professionals with medical students.

Please, clarify the abbreviation “NTC”.

What the rationale of exclusion of the 4th year of medical school was? Please, specify.

I suggest to move the obtained Cronbach’s alpha value in the results section.

I suggest to delete the Bias paragraph: it is redundant since data recording is described in the Data analysis and the attempt to minimize selection bias in the Data Collection procedure.

I suggest to replace “year of graduation” with “year of medical study”.

The selection of the participants, as indicated in the Methods, reflects a classic example of a convenience sampling. Hence, the sample could be biased and not a truly random sample. This is a major limitation that undermines the validity of this study and suggests that the obtained sample could not be representative. The selection bias would have affected the external validity in terms of generalizing the findings to the wider population, and I am not sure if the attempt of the authors to minimize the selection bias was enough.

The authors should mention whether or not an incentive was offered for completion of the questionnaire.

Statistical analysis

No information is given about non-responding, if any.

Results

Please, clarify what the response rate was.

To avoid misunderstanding, I suggest to clarify what “RR” is. I suppose it refers to a ratio between two prevalences to analyse the association since the design of the study is cross-sectional (rate ratio), but it could be confused with a "relative risk".

The p- value of gender and year of study in the description of model 1 have been repeated twice. It is unnecessary p-value if the CIs were reported.

Discussion

The authors do not properly delineate discussion. The discussion is to analyze the meaning of the results, and to make them meaningful. It seems that the literature review to generate a well structured discussion and better define the theoretical framework of the research was not deep enough.

The study findings showed that male gender is a factor associated with knowledge gaps among medical students. I suggest, to comment the role of parents of male adolescents and acceptability of vaccination also among young males. The following paper have to be cited and commented: Hum Vaccin Immunother. 2014;10(9):2536-42, Hum Vaccin Immunother. 2016 Jun 2;12(6):1504-10, Hum Vaccin Immunother. 2016;12(1):47-51.

I suggest to add a comment about potential determinants of “vaccine hesitancy”, e.g. according to the WHO Strategic Advisory Group of Experts (SAGE) on Immunization. The following studies have to be cited and commented: Bianco et al. Vaccine 2019;37:984-999 and Napolitano et al. Hum Vaccin Immunother. 2018 Jul 3;14(7):1558-1565.

The authors should address the pivotal role of healthcare providers in promoting HPV vaccination in different setting and in specific risk groups. I suggest to cite and comment other studies (e.g. Napolitano F, et al. PLoS One. 2018 Mar 29;13(3):e0194920; Landis K et al. Vaccine. 2018 Jun 7;36(24):3498-3504; D’Alessandro et al. Hum Vaccin Immunother 2018;14:1573-1579; etc..).

In the limitations of the study, the authors did not mention the main limits of the cross-sectional survey.

In all Tables, please include only one decimal number.

Reviewer #2: The author provided an important topic for the paper; however, there are concerns in the manuscript.

1.Study title needs to add setting (Brazil). Title and study purposes need to revise based on study variables. For example, sources of HPV information were also explored in this study.

2.Introduction: Literature review needs to be more focused and related to the study variables and population. Please add HPV vaccination guidelines and insurance coverage in Brazil. Please add prevalence and mortality data of cervical cancer and other HPV-related diseases among men and women in Brazil.

3.Method: make eligibility and non-inclusion criteria more concise. Eligibility needs to add age criteria. Instrument- use table to list all 31 questions. It’s not clear about why 80% of correct answers was used to determine “adequate.”

4.Results: please clarify why odds ratios were performed in Table 1.

5.Discussion: Need to add more citations related to evidence-based practice and policy recommendations. Please also add study settings and location for citations.

6. PLOS authors have the option to publish the peer review history of their article (what does this mean?). If published, this will include your full peer review and any attached files.

Reviewer #1: No

Reviewer #2: No

---

## [Author Response · Author response to Decision Letter 0]

6 Feb 2020

Dear Editor,

We are grateful for the considerations about our manuscript, "Knowledge Gaps and Acquisition About HPV and its Quadrivalent Vaccine Among Medical Students" (Manuscript ID: PONE-D-19-33466), to PLOS ONE.

This response letter contains a point-by-point reply to the peer reviewer’s comments, outlining the changes we have made. 

Reviewer 1

Comment 1: 

- Title

“I suggest to delete in the title and in the introduction the word “quadrivalent”. The survey questions generically explored HPV vaccine.”

Answer to Comment 1: We kindly appreciate the suggestion and made the necessary changes to the manuscript:

“Knowledge gaps and acquisition about HPV and its vaccine among medical students”

Comment 2:

- Introduction 

“ In the first paragraph, the authors stated “The skills and competences acquired by these students will be used for women's health care…” Long-lasting infections with high-risk HPVs can cause cancer in different parts of the body, such as oropharynx, anus, rectum, penis, and not only in the cervix, vagina, and vulva. In developed countries, high-risk HPVs cause 3% of all cancers in women and 2% of all cancers in men. Please, clarify that the competences acquired by medical students will be also useful to protect men's health care.”

Answer to Comment 2: We kindly appreciate the suggestion and made the necessary changes to the manuscript:

“…The skills and competences acquired by these students will be used for men's and women's health care, especially for HPV-induced cancer (cervical, oropharynx, anus, rectum and penis) and precursor lesions at different levels of care, including health counseling and education, prevention, diagnosis, treatment and recovery”

Comment 3: 

- Introduction 

“I suggest to add a brief description of what the Brasil’s National Immunization Program recommend regarding HPV vaccination.”

Answer to Comment 3: We kindly appreciate the suggestion and made the necessary changes to the manuscript:

“As future health care providers, these medical students will perform preventive actions (HPV vaccination) and promotion of cervical cancer screening4, which is a high prevalence and mortality disease in Brazil and worldwide, representing a public health problem. In Brazil the HPV vaccine is available in the National Immunization Program for female population aged nine to 14 years old, male population aged 11 to 14 years old.”

Comment 4: 

- Method

“The authors described how the sample size was calculated, but the reference used for sample size calculation to estimate the prevalence was not included.”

Answer to Comment 4: We agreed and made the inclusion

Reference: Agranonik, M., & Hirakata, V. N. (2011). Cálculo de tamanho de amostra: proporções. Clinical & Biomedical Research, 31(3).

Comment 5: 

- Method

“It would seem that “did not accept” to participate in the study was an ineligibility criterion, but the number of subjects who refused to answer the survey, represents a fraction of people in the total sample.”

Answer to Comment 5: We kindly appreciate the suggestion

We considered inclusion only the student who agreed to participate in the research or signed the consent form and answered the questionnaire. There were no individuals who signed the term and did not respond the questionnaire. 

Please consider to check our response rate per student on method. 

Comment 6: 

- Method

“In the Instrument paragraph, no. 6), and throughout the manuscript, I suggest to replace health professionals with medical students.”

Answer to comment 6: We kindly appreciate the suggestion and made the necessary changes to the manuscript. 

Comment 7: 

- Method

“Please, clarify the abbreviation “NTC”.”

Answer to Comment 7: We kindly appreciate the suggestion and made the necessary changes to the manuscript:

We replaced “NTC” with “NS (not sure).

Comment 8: 

- Method

“What the rationale of exclusion of the 4th year of medical school was? Please, specify.”

Answer to Comment 8: We kindly appreciate the suggestion.

The main reason was to eliminate bias, as students in the fourth year of medical study are considered transitional, with knowledge acquired from basic cycle and also already acts as an senior student.

Comment 9: 

- Method

“I suggest to move the obtained Cronbach’s alpha value in the results section.”

Answer to Comment 9: We kindly appreciate the suggestion and made the necessary changes to the manuscript. 

Comment 10: 

- Method

“I suggest to delete the Bias paragraph: it is redundant since data recording is described in the Data analysis and the attempt to minimize selection bias in the Data Collection procedure.”

Answer to Comment 10: We kindly appreciate the suggestion and made the necessary changes to the manuscript. 

Comment 11: 

- Method

“I suggest to replace “year of graduation” with “year of medical study”.”

Answer to Comment 11: We kindly appreciate the suggestion and made the necessary changes to the manuscript:

We replaced “year of graduation” with “year of medical study”.

Comment 12: 

- Method

“The selection of the participants, as indicated in the Methods, reflects a classic example of a convenience sampling. Hence, the sample could be biased and not a truly random sample. This is a major limitation that undermines the validity of this study and suggests that the obtained sample could not be representative. The selection bias would have affected the external validity in terms of generalizing the findings to the wider population, and I am not sure if the attempt of the authors to minimize the selection bias was enough.”

Answer to Comment 12: We kindly appreciate the suggestion.

We consider our convenience sample as representative because it is a closed and known population. We included only students enrolled and attending a known and specific school curriculum. Still, the questionnaire was not applied in the disciplines that talked about the theme. The questionnaire was applied at the convenience and availability of the interviewers and interviewees, it was not scheduled. The interviewers were students, that is, there was no presence of the Professor that could intimidate or induce the students' response.

Under these conditions the convenience sample was applied and this will be included as a limitation in the discussion.

“Limitations of the study include the cross-sectional design which does not allow understanding over time how and when individuals gained knowledge about HPV. The instrument did identify other possible associated factors such as ethnicity, culture and religion that may be associated. The data collection took place in an internationally recognized University by convenience sample and findings may not be representative of other Brazilian medical schools.”

Comment 13: 

- Method

“The authors should mention whether or not an incentive was offered for completion of the questionnaire.”

Answer to comment 13: We kindly appreciate the suggestion and made the necessary changes to the manuscript. 

“A convenience sample of students from the first to sixth year of medical school were invited to participate voluntarily in the research whit no financial incentive offered to the respondents.”

Comment 14: 

- Method, Statistical analysis

“No information is given about non-responding, if any.”

Answer to comment 14: We kindly appreciate the suggestion and made the necessary changes to the manuscript. 

“Non-responding were represented as missing and did not exceed 5% of question-by-question responses.”

Comment 15: 

- Results

“Please, clarify what the response rate was.”

Answer to comment 15: We kindly appreciate the suggestion and made the necessary changes to the manuscript. 

“The response rate of the variables (questionnaire questions) was an average of 99.2% (SD 0.9%).”

Comment 16:

- Results

“To avoid misunderstanding, I suggest to clarify what “RR” is. I suppose it refers to a ratio between two prevalences to analyse the association since the design of the study is cross-sectional (rate ratio), but it could be confused with a "relative risk".”

Answer to comment 16: We kindly appreciate the suggestion and made the necessary changes to the manuscript. In fact, the “RR” represents the prevalence ratio and we will insert this into the statistical analysis in the method. We replaced relative risk (RR) for prevalence rate (PR).

Comment 17:

- Results

“The p- value of gender and year of study in the description of model 1 have been repeated twice. It is unnecessary p-value if the CIs were reported.”

Answer to comment 17: We kindly appreciate the suggestion and made the necessary changes to the manuscript. We agree and we will report only the CIs at supplementary material. 

Comment 18:

- Discussion

“The authors do not properly delineate discussion. The discussion is to analyze the meaning of the results, and to make them meaningful. It seems that the literature review to generate a well structured discussion and better define the theoretical framework of the research was not deep enough.”

Answer to comment 18: 

we appreciate the comment. The authors included new references and references suggested by the reviewers listed above: 

References:

1- Cinar O, Ozkan S, Aslan GK, Alatas E. Knowledge and Behavior of University Students toward Human Papillomavirus and Vaccination. Asia Pac J Oncol Nurs. 2019;6:300-7.

2- Bianco A, Pileggi C, Iozzo F, Nobile CG, Pavia M. Vaccination against human papilloma virus infection in male adolescents: knowledge, attitudes, and acceptability among parents in Italy. Hum Vaccin Immunother. 2014;10(9):2536-42.

3- Napolitano F, Napolitano P, Liguori G, Angelillo IF. Human papillomavirus infection and vaccination: Knowledge and attitudes among young males in Italy. Hum Vaccin Immunother. 2016 Jun 2;12(6):1504-10.

Comment 19:

- Discussion

“The study findings showed that male gender is a factor associated with knowledge gaps among medical students. I suggest, to comment the role of parents of male adolescents and acceptability of vaccination also among young males. The following paper have to be cited and commented: Hum Vaccin Immunother. 2014;10(9):2536-42, Hum Vaccin Immunother. 2016 Jun 2;12(6):1504-10, Hum Vaccin Immunother. 2016;12(1):47-51.”

Answer to comment 19: We kindly appreciate the suggestion and made the necessary changes to the manuscript:

“While the higher level of knowledge found in females in this study is a function of the personal and individual experiences, since medical students of both sexes are exposed to the same amount of theoretical concepts before graduation1. In addition, a study shows that male parents underestimate the effects of HPV in males and prioritize the vaccine for women2. It is necessary to disseminate correct information to parents and to emphasize the fact that they are not actively involved and open to dialogue when it comes to their children's sexual education3.”

References:

1- Cinar O, Ozkan S, Aslan GK, Alatas E. Knowledge and Behavior of University Students toward Human Papillomavirus and Vaccination. Asia Pac J Oncol Nurs. 2019;6:300-7.

2- Bianco A, Pileggi C, Iozzo F, Nobile CG, Pavia M. Vaccination against human papilloma virus infection in male adolescents: knowledge, attitudes, and acceptability among parents in Italy. Hum Vaccin Immunother. 2014;10(9):2536-42.

3- Napolitano F, Napolitano P, Liguori G, Angelillo IF. Human papillomavirus infection and vaccination: Knowledge and attitudes among young males in Italy. Hum Vaccin Immunother. 2016 Jun 2;12(6):1504-10.

Comment 20:

- Discussion

“I suggest to add a comment about potential determinants of “vaccine hesitancy”, e.g. according to the WHO Strategic Advisory Group of Experts (SAGE) on Immunization. The following studies have to be cited and commented: Bianco et al. Vaccine 2019;37:984-999 and Napolitano et al. Hum Vaccin Immunother. 2018 Jul 3;14(7):1558-1565.”

Answer to comment 20: We kindly appreciate the suggestion and made the necessary changes to the manuscript:

“It is worth mentioning that the phenomenon called vaccine hesitation (which consists of delay in accepting or refusing vaccines, despite the availability of vaccination services)1, is a fundamental determinant for low HPV vaccination rates. Studies show that potential determinants of vaccine hesitation are communication and preventive attitudes. Health professionals act as a key factor in reducing vaccine hesitation2,3.”

References:

1- The Lancet Child Adolescent Health. Vaccine hesitancy: a generation at risk. Lancet Child Adolesc Health. 2019 May;3(5):281.

2- Bianco A, Mascaro V, Zucco R, Pavia M. Parent perspectives on childhood vaccination: How to deal with vaccine hesitancy and refusal? Vaccine. 2019 Feb 8;37(7):984-990. doi: 10.1016/j.vaccine.2018.12.062. Epub 2019 Jan 14.

3- Napolitano F, D'Alessandro A, Angelillo IF. Investigating Italian parents' vaccine hesitancy: A cross-sectional survey. Hum Vaccin Immunother. 2018 Jul;14(7):1558-1565.

Comment 21:

- Discussion

“The authors should address the pivotal role of healthcare providers in promoting HPV vaccination in different setting and in specific risk groups. I suggest to cite and comment other studies (e.g. Napolitano F, et al. PLoS One. 2018 Mar 29;13(3):e0194920; Landis K et al. Vaccine. 2018 Jun 7;36(24):3498-3504; D’Alessandro et al. Hum Vaccin Immunother 2018;14:1573-1579; etc..).”

Answer to comment 21:

We appreciated your comments and included the following paragraph: 

“It is worth mentioning that the phenomenon called vaccine hesitation (which consists of delay in accepting or refusing vaccines, despite the availability of vaccination services), is a fundamental determinant for low HPV vaccination rates. Studies show that potential determinants of vaccine hesitation are communication and preventive attitudes. Health professionals act as a key factor in reducing vaccine hesitation.

Studies demonstrate the importance of the physician in primary health care setting and vaccination counseling. Furthermore, the authors describe care practices based on the patient's medical relationship and health promotion interventions based on the individual care. Our study shows information and knowledge gaps that must be acquired during the medical graduation so that the student can exercise health education and individual counseling of his patients.”

Comment 22:

- Discussion

“In the limitations of the study, the authors did not mention the main limits of the cross-sectional survey.”

Answer to comment 22: We agree and we will include a comment at discussion. 

Limitations of the study include the cross-sectional design which does not allow understanding over time how and when individuals gained knowledge about HPV, it is not possible to distinguish whether knowledge acquisition was a cause or a consequence according to the student's level of education, neither the individual acquisition of each student.

Comment 23:

- Discussion

“In all Tables, please include only one decimal number.”

Answer to comment 23: We kindly appreciate the suggestion and made the necessary changes to the manuscript.

Reviewer 2

Comment 1:

- Title

“Study title needs to add setting (Brazil). Title and study purposes need to revise based on study variables. For example, sources of HPV information were also explored in this study.”

Answer to comment 1:

We kindly appreciate the suggestion and the changes were made according to recommendations of all reviewers: “KNOWLEDGE GAPS AND ACQUISITION ABOUT HPV AND ITS VACCINE AMONG BRAZILIAN MEDICAL STUDENTS”

Comment 2:

- Introduction

“Literature review needs to be more focused and related to the study variables and population. Please add HPV vaccination guidelines and insurance coverage in Brazil. Please add prevalence and mortality data of cervical cancer and other HPV-related diseases among men and women in Brazil.”

Answer to comment 2: 

We appreciated your comments and included the following paragraph:

“The skills and competences acquired by these students will be used for men's and women's health care, especially for HPV-induced cancer (cervical, oropharynx, anus, rectum and penis) and precursor lesions at different levels of care, including health counseling and education, prevention, diagnosis, treatment and recovery2,3.”

Comment 3:

- Method

“Make eligibility and non-inclusion criteria more concise. Eligibility needs to add age criteria. Instrument- use table to list all 31 questions. It’s not clear about why 80% of correct answers was used to determine “adequate”.”

Answer to comment 3: 

We appreciated your comments and resumed the eligibility and non-inclusion criteria paragraphs as follow:

“…A convenience sample of both sexes students over 18 years old from the first to sixth year of medical school were invited to participate voluntarily in the research, whit no financial incentive offered to the respondents, who signed the Informed Consent Form.

Non-inclusion criteria: Undergraduate students with cognitive problems or any interest that influences the answer to the questions contained in the data collection instrument.”

Furthermore the 31 questions were presented in a table.

We apologize and we clarify:

The good level of knowledge was classified as adequate when the percentage of correct answers exceeded 80% and was based on the previously published score in a similar population at reference:

Pereira JEG, Gomes JM, Costa AS, Figueiredo FWDS, Adami F, Santos EFS, Sorpreso ICE, Abreu LC. Knowledge and acceptability of the human papillomavirus vaccine among health professionals in Acre state, western Amazon. Clinics (Sao Paulo). 2019;74:e1166. 

Comment 4:

- Results

“Please clarify why odds ratios were performed in Table 1.”

Answer to comment 4: 

We apologize, the odds ratio was not used in Table 1 or in our study. The prevalence ratio was described in tables 2, 3 and supplementary material. The changes were made according to the comments of the other reviewers.

Comment 5:

- Discussion

“Need to add more citations related to evidence-based practice and policy recommendations. Please also add study settings and location for citations.”

Answer to comment 5:

We appreciated your comments and included the following paragraphs:

“Programs for successful implementation of HPV vaccination include factors such as intersectoral involvement (across the education and health sector), as well as collaboration between institutions in the health, education and financial sectors. The future efforts should focus on programs that can be implemented within health care settings, such as reminder and recall strategies and physician-focused efforts, as well as the use of alternative community-based locations, such as schools.

In Brazil, in 2014, vaccine implementation had been used together with the education sector in public and private schools, bringing benefits in terms of vaccine coverage rates. Our study reinforces the themes that must be addressed during medical training in order to strengthen the physician's action in providing information and patient’s counseling.” 

We added the references below:

1) Niccolai LM, Hansen CE. Practice- and Community-Based Interventions to Increase Human Papillomavirus Vaccine Coverage: A Systematic Review. JAMA Pediatr. 2015;169(7):686-92.

2) Brotherton JML, Bloem PN. Population-based HPV vaccination programmes are safe and effective: 2017 update and the impetus for achieving better global coverage. Best Pract Res Clin Obstet Gynaecol. 2018;47:42-58. 

3) Baker ML, Figueroa-Downing D, Chiang ED, Villa L, Baggio ML, Eluf-Neto J, Bednarczyk RA, Evans DP. Paving pathways: Brazil's implementation of a national human papillomavirus immunization campaign. Rev Panam Salud Publica. 2015;38(2):163-6. 

Sincerely,

Isabel Cristina Esposito Sorpreso

Professor Doutor

Disciplina de Ginecologia

Departamento de Obstetrícia e Ginecologia

Faculdade de Medicina da Universidade de São Paulo

CV: http://lattes.cnpq.br/9672065408641518

ORCID: 0000-0002-5475-5957

---

## [Decision Letter · Decision Letter 1]

18 Feb 2020

PONE-D-19-33466R1

KNOWLEDGE GAPS AND ACQUISITION ABOUT HPV AND ITS VACCINE AMONG BRAZILIAN MEDICAL STUDENTS

PLOS ONE

Dear Esposito Sorpreso,

Thank you for submitting your manuscript to PLOS ONE. After careful consideration, we feel that it has merit but does not fully meet PLOS ONE’s publication criteria as it currently stands. Therefore, we invite you to submit a revised version of the manuscript that addresses the points raised during the review process.

We would appreciate receiving your revised manuscript by February 29, 2020. To enhance the reproducibility of your results, we recommend that if applicable you deposit your laboratory protocols in protocols.io, where a protocol can be assigned its own identifier (DOI) such that it can be cited independently in the future. For instructions see: http://journals.plos.org/plosone/s/submission-guidelines#loc-laboratory-protocols

We look forward to receiving your revised manuscript.

Kind regards,

Italo Francesco Angelillo, DDS, MPH

Academic Editor

PLOS ONE

Reviewers' comments:

Reviewer's Responses to Questions

**Comments to the Author**

1. If the authors have adequately addressed your comments raised in a previous round of review and you feel that this manuscript is now acceptable for publication, you may indicate that here to bypass the “Comments to the Author” section, enter your conflict of interest statement in the “Confidential to Editor” section, and submit your "Accept" recommendation.

Reviewer #1: All comments have been addressed

Reviewer #3: (No Response)

2. Is the manuscript technically sound, and do the data support the conclusions?

Reviewer #1: Yes

Reviewer #3: Yes

3. Has the statistical analysis been performed appropriately and rigorously? 

Reviewer #1: Yes

Reviewer #3: Yes

4. Have the authors made all data underlying the findings in their manuscript fully available?

Reviewer #1: Yes

Reviewer #3: Yes

5. Is the manuscript presented in an intelligible fashion and written in standard English?

Reviewer #1: Yes

Reviewer #3: Yes

6. Review Comments to the Author

Reviewer #1: (No Response)

Reviewer #3: The topic addressed in the manuscript is important in terms of public health. It seems to me that the authors addressed properly all comments raised by the previous reviewers. The paper may be published and I have only some minor comments.

1) In the abstract, the Cronbach's alpha value of 0.74 should be reported in the results section.

2) In the results section the paragraph concerning the internal validity of the questionnaire should be at the beginning and not at the end of the section.

3) There is an error in the reporting of 95% CI of the PR concerning the year of study, both in the abstract and in the result section.

7. PLOS authors have the option to publish the peer review history of their article (what does this mean?). If published, this will include your full peer review and any attached files.

Reviewer #1: No

Reviewer #3: No

---

## [Author Response · Author response to Decision Letter 1]

19 Feb 2020

Dear Reviewers,

We are grateful for the considerations about our manuscript, "Knowledge Gaps and Acquisition About HPV and its Quadrivalent Vaccine Among Medical Students" (Manuscript ID: PONE-D-19-33466), to PLOS ONE.

This response letter contains a point-by-point reply to the peer reviewer’s comments, outlining the changes we have made. 

Reviewer #1

Comment 1: No comments

Reviewer #3

The topic addressed in the manuscript is important in terms of public health. It seems to me that the authors addressed properly all comments raised by the previous reviewers. The paper may be published and I have only some minor comments.

Comment 1:

“In the abstract, the Cronbach's alpha value of 0.74 should be reported in the results section.”

Answer to Comment 1: We kindly appreciate the suggestion and made the necessary changes in the abstract:

“Results: To evaluate the internal consistency of the instrument applied, the Cronbach's alpha equation was used, obtaining the alpha value (α) = 0.74 for this population. This value attests that the consistency of the answers obtained with this questionnaire is considered substantial and acceptable.”

Comment 2:

“In the results section the paragraph concerning the internal validity of the questionnaire should be at the beginning and not at the end of the section.”

Answer to Comment 2: We kindly appreciate the suggestion and put the paragraph concerning the internal validity of the questionnaire at the beginning of the section Results.

Comment 3:

“There is an error in the reporting of 95% CI of the PR concerning the year of study, both in the abstract and in the result section.”

Answer to Comment 3: We kindly appreciate the correction and made the necessary changes in the abstract and in the results section:

Abstract: “Students in the first, second and third year of study had a 51% higher risk of a knowledge gap when compared to students in the final years of graduation [PR 1.51 (1.3:1.8); p <0.001].”

Results: “It is observed that there was a difference in students by grade with first, second and thrid year students having a 51% risk of having a knowledge gap than students in the final cycle [PR 1.51 (1.3:1.8)]”

Sincerely,

Isabel Cristina Esposito Sorpreso

Professor Doutor

Disciplina de Ginecologia

Departamento de Obstetrícia e Ginecologia

Faculdade de Medicina da Universidade de São Paulo

CV: http://lattes.cnpq.br/9672065408641518

ORCID: 0000-0002-5475-5957

---

## [Editor Report · Decision Letter 2]

21 Feb 2020

KNOWLEDGE GAPS AND ACQUISITION ABOUT HPV AND ITS VACCINE AMONG BRAZILIAN MEDICAL STUDENTS

PONE-D-19-33466R2

Dear Dr. Esposito,

We are pleased to inform you that your manuscript has been judged scientifically suitable for publication and will be formally accepted for publication once it complies with all outstanding technical requirements.

With kind regards,

Italo Francesco Angelillo, DDS, MPH

Academic Editor

PLOS ONE
---

## [Editor Report · Acceptance letter]

5 Mar 2020

PONE-D-19-33466R2 

Knowledge Gaps and Acquisition about HPV and Its Vaccine Among Brazilian Medical Students 

Dear Dr. Sorpreso:

I am pleased to inform you that your manuscript has been deemed suitable for publication in PLOS ONE. Congratulations! Your manuscript is now with our production department. 

With kind regards,

on behalf of

Professor Italo Francesco Angelillo 

Academic Editor

PLOS ONE